**Data Availability Statement:** All relevant data are within the manuscript and its Supporting Information files.

**Funding:** The author(s) received no specific funding for this work.

# Impact of social isolation due to COVID-19 on the seasonality of pediatric respiratory diseases

**Milena Siciliano Nascimento**  **\*, Diana Milena Baggio, Linus Pauling Fascina, Cristiane do Prado**

Department of Pediatrics, Hospital Israelita Albert Einstein, São Paulo, Brazil

\* milenasn@einstein.br

## Abstract

### Introduction

Respiratory tract diseases are the major cause of morbidity and mortality in children under the age of 5 years, constituting the highest rate of hospitalization in this age group.

### Objectives

To determine the prevalence of hospitalizations for respiratory diseases in childhood in the last 5 years and to assess the impact of social isolation due to COVID-19 on the seasonal behavior of these diseases.

### Methods

A cross-sectional clinical study was carried out, with a survey of all patients aged 0 to 17 years who were admitted with a diagnosis of respiratory diseases between January 2015 and July 2020. The database was delivered to the researchers anonymized. The variables used for analysis were date of admission, date of discharge, length of stay, age, sex and diagnosis. In order to make the analysis possible, the diagnoses were grouped into upper respiratory infection (URI), asthma / bronchitis, bronchiolitis and pneumonia.

### Results

2236 admissions were included in the study. Children under 5 years old account for 81% of hospitalizations for respiratory disease in our population. In the adjusted model, an average reduction of 38 hospitalizations was observed in the period of social isolation (coefficient: -37.66; 95% CI (- 68.17; -7.15); p = 0.016).

### Conclusion

The social isolation measures adopted during the COVID-19 pandemic dramatically interfered with the seasonality of childhood respiratory diseases. This was reflected in the unexpected reduction in the number of hospitalizations in the pediatric population during this period.

**Competing interests:** The authors have declared that no competing interests exist.

## Introduction

Respiratory tract diseases are the major cause of morbidity and mortality in children under the age of 5 years, constituting the highest rate of hospitalization in this age group [1, 2]. In developing countries, diseases of the lower respiratory tract represent about 90% of deaths from respiratory diseases, most of which are bronchial and alveolar infections of viral origin [1, 3]. Regarding upper respiratory infections (URI), infants and preschoolers develop, on average, six to eight infections per year [4]. This infection profile, despite not causing serious illness, is responsible for epidemics due to the continuous circulation of pathogens in the community [5].

In this context, it is important to highlight the role of day-care centers and schools that start in increasingly younger age groups and several studies point out as being an important risk factor for the acquisition of respiratory infections, especially in children from zero to two years of age, due to greater children's exposure to infectious agents, confinement and agglomeration. In addition, children can act as sources of infection in their families, further spreading infectious agents in the community [5, 6].

In the year 2020, due to the pandemic caused by COVID-19, social isolation, with the closing of trade and companies and the suspension of classes, was highly recommended in an attempt to reduce the spread of the virus. The impact caused by social isolation was one of the factors that may have contributed to the change in the seasonal characteristic of respiratory diseases in pediatrics in the year in question [7].

In Brazil, social isolation measures that were included suspending classes at schools and universities, closing companies and commerce and banning events, have been started in mid-March. The easing of these measures began in mid-July, however, schools and day care remain closed [8].

The present study aims to determine the incidence of hospitalizations for respiratory diseases in childhood in the last 5 years and to assess the impact of social isolation due to COVID-19 on the seasonal behavior of these diseases.

## Methods

### Study type and location

A cross-sectional study was carried out by collecting epidemiological data on hospital admissions for respiratory diseases in pediatric patients, at private hospital from January 2015 to July 2020. Patients aged 0 to 17 years were included, who needed of hospitalization with diagnoses of respiratory diseases. Our institution is a tertiary level private hospital that providence approximately 3,000 admissions per year

### Protocol

After approval of the project by the Research Ethics Committee of the Hospital Israelita Albert Einstein, a survey was made of all patients aged 0 to 17 years who were hospitalized with primary and secondary diagnosis of respiratory diseases (International Disease Code, 10th Revision: J00 –J99) from January 2015 to July 2020. The database was delivered to the researchers anonymously and, for this reason, the informed consent term was waived by the ethics and research committee.

The variables used for analysis were date of admission, date of discharge, length of stay, age, sex and diagnosis. To make the analysis possible, the diagnoses were grouped using the subcategories by similarity criteria into upper airway infection (URI) asthma / bronchitis, bronchiolitis and pneumonia. The exact grouping of the ICD-10 codes into each subcategory was

specified in Table 1. Asthma and bronchitis were grouped because diagnosis of bronchitis was closely associated with the secondary diagnosis of asthma.

## Statistical analysis

The data were described by means of absolute and relative frequencies for categorical variables and by mean values and standard deviations (SD), minimum, maximum and quartiles for numerical variables. The distributions of continuous variables were investigated using box-plots and histograms.

Generalized linear models with Gamma distribution and logarithmic link function were adjusted to estimate the length of hospital stay and to investigate associations with periods of social isolation Categorical variables were compared between groups with or without social isolation due to COVID-19 using chi-square or Fisher's exact tests

The number of hospitalizations over time and their relationship with social isolation was assessed by a time series regression model, with ARIMA identification. The model considered

**Table 1. ICD-10 codes into each subcategory.**

| Upper airway infection (UAI) | J02.8 Acute pharyngitis due to other specified organisms |
|---|---|
| | J02.9 Acute pharyngitis, unspecified |
| | J04.0 Acute laryngitis |
| | J04.1 Acute tracheitis |
| | J04.2 Acute laryngotracheitis |
| | J06.9 Acute upper respiratory infection, unspecified |
| | J38.6 Stenosis of larynx |
| | J39.2 Other diseases of pharynx |
| Asthma /Bronchitis | J45.1 Nonallergic asthma |
| | J45.8 Mixed asthma |
| | J45.9 Other and unspecified asthma |
| | J20.0 Acute bronchitis due to Mycoplasma pneumoniae |
| | J20.5 Acute bronchitis due to respiratory syncytial virus |
| | J20.6 Acute bronchitis due to rhinovirus |
| | J20.8 Acute bronchitis due to other specified organisms |
| | J20.9 Acute bronchitis, unspecified |
| | J21.9 Acute bronchiolitis, unspecified |
| | J40 Bronchitis, not specified as acute or chronic |
| Bronchiolitis | J21.0 Acute bronchiolitis due to respiratory syncytial virus |
| | J21.8 Acute bronchiolitis due to other specified organisms |
| Pneumonia | J10.1 Influenza due to other identified influenza virus with other respiratory manifestations |
| | J12.0 Adenoviral pneumonia |
| | J12.1 Respiratory syncytial virus pneumonia |
| | J12.2 Parainfluenza virus pneumonia |
| | J12.8 Other viral pneumonia |
| | J15.7 Pneumonia due to Mycoplasma pneumoniae |
| | J15.8 Pneumonia due to other specified bacteria |
| | J15.9 Unspecified bacterial pneumonia |
| | J17 Pneumonia in diseases classified elsewhere |
| | J18.0 Bronchopneumonia, unspecified organism |
| | J18.9 Pneumonia, unspecified organism |

the monthly numbers of hospitalizations for respiratory diseases in pediatric patients in the first semester and social isolation in the period from April to June 2020 was used as an explanatory variable, with the coefficient and p-value for the absence of alteration test being presented. and the level of significance adopted was 5%.

The analyzes were performed with the aid of the SPSS program (version 24.0) and the R (version 4.0.2) and the additional packages: forecast, Rcmdr, RcmdrPlugin.EZR and lmtest. For all analyzes, the 5% significance level was adopted.

## Results

The database contained 2618 records and hospitalizations, of which 382 were excluded, as they had primary and secondary diagnoses that did not meet the study's inclusion criteria. Thus, 2236 admissions were included in the study.

The main demographic characteristics as well as the presentation of the main diagnoses are shown in Table 2.

The main demographic characteristics as well as the presentation of the main diagnoses are shown in Table 1. This table also shows the comparisons between period with (April/2020 – June/2020) or without social isolation (January/2015 –march/2020). Children under 5 years old accounted for 81.3% of hospitalizations for respiratory disease in the period without social isolation. During social isolation we observed a significant reduction, with children under 5

**Table 2. Characteristics of pediatric patients admitted for respiratory diseases in the period from January 2015 to June 2020 (n = 2236).**

|  | Total (2236) | Social isolation due to COVID-19 | | p-value |
|---|---|---|---|---|
|  |  | No (n = 2216) | Yes (n = 20) |  |
| Sex |  |  |  | 0.571 [1] |
| Female | 1089 (48.7%) | 1078 (48.6%) | 11 (55.0%) |  |
| Male | 1147 (51.3%) | 1138 (51.4%) | 9 (45.0%) |  |
| Age group |  |  |  | 0.002 [2] |
| 0 to 2 years old | 1267 (56.7%) | 1260 (56.9%) | 7 (35.0%) |  |
| 3 to 5 years old | 543 (24.3%) | 541 (24.4%) | 2 (10.0%) |  |
| 6 to 10 years old | 331 (14.8%) | 322 (14.5%) | 9 (45.0%) |  |
| 11 to 17 years old | 95 (4.2%) | 93 (4.2%) | 2 (10.0%) |  |
| Diagnosis based on CID* |  |  |  | 0.153 [2] |
| Asthma / Bronchitis | 223 (10.0%) | 221 (10.0%) | 2 (10.5%) |  |
| Bronchiolitis | 163 (7.2%) | 163 (7.3%) | 0 (0.0%) |  |
| UAI (upper airway infection) | 249 (11.1%) | 244 (11.0%) | 5 (26.3%) |  |
| Pneumonia | 1601 (71.6%) | 1589 (71.7%) | 13 (63.2%) |  |
| Infection Agent |  |  |  | 0.141 [1] |
| Bacterium | 1466 (65.6%) | 1456 (65.7%) | 10 (50.0%) |  |
| Virus | 770 (34.4%) | 760 (34.3%) | 10 (50.0%) |  |
| Length of hospital stay (days) # |  |  |  |  |
| Mean (IC 95%) | 4.21 (4.09; 4.34) | 4.23 (4.10; 4.35) | 2.70 (1.98; 3.69) | 0.005 [3] |
| Minimum; Maximum | 0.13; 112.00 | 0.13; 112.00 | 1.00; 6.00 |  |

Without social isolation: january/2015 –march/2020; with social isolation: april/2020 –june/2020

p-values for the chi-square test(1), Fisher's exact test(2) and generalized linear model (3).

#: mean and 95% confidence interval estimated by generalized linear model. For those with less than one day in the hospital, we assumed three hours of stay, which equates to 0,125 days of hospitalization

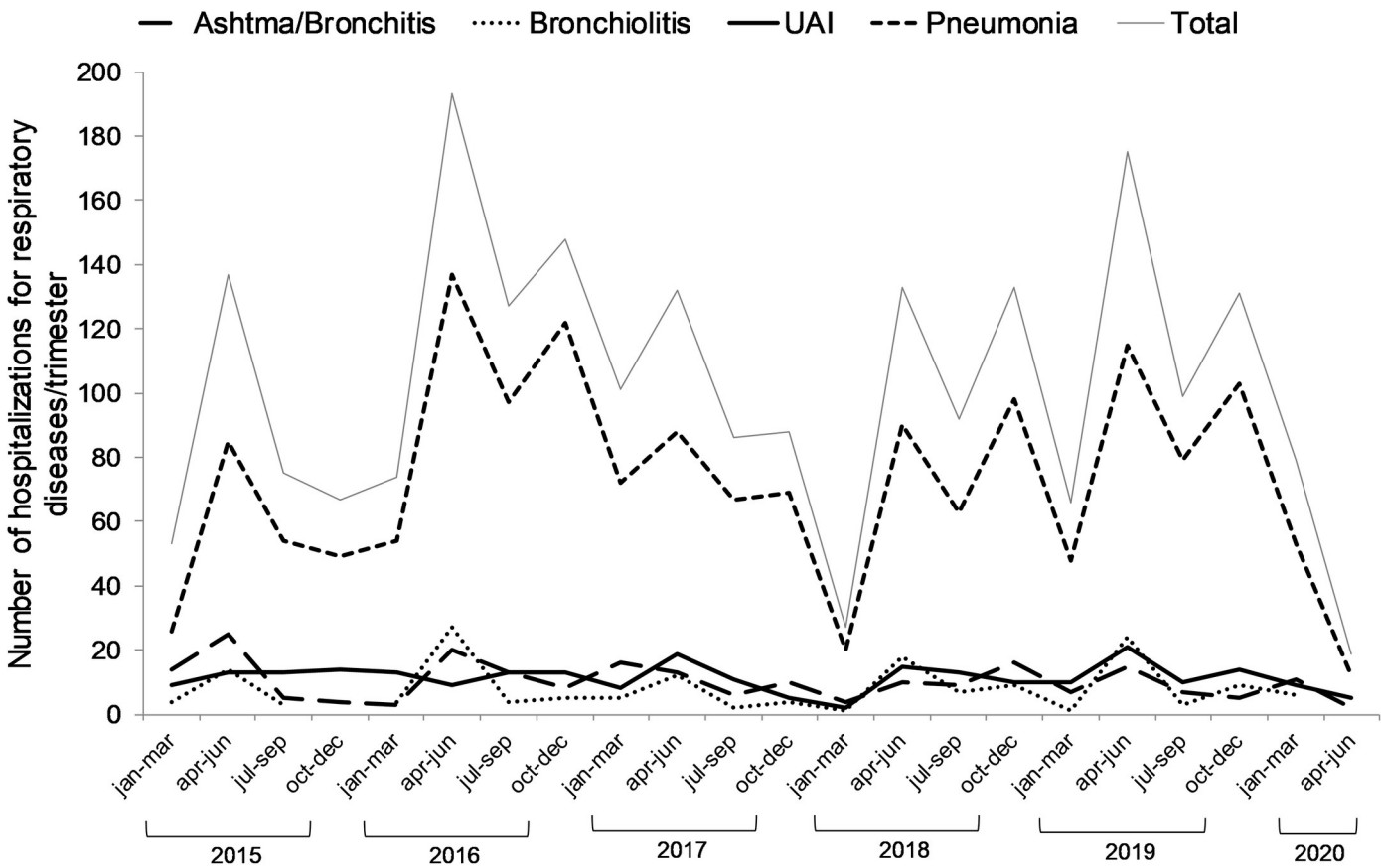

**Fig 1. Distribution of the number of hospitalizations for respiratory diseases in pediatric patients from January 2015 to June 2020 total and according to diagnosis (n = 2236).**

years old representing 45% of hospitalizations (p-value = 0.002). We also observed a reduction in the length of hospital stay (p-value = 0,005).

The distribution of the number of hospitalizations per trimester in the period from 2015 to 2020, total and according to diagnosis is shown in Fig 1.

In the ARIMA model, we observed in the period with social isolation (April to June 2020) an average reduction of 38 hospitalizations for respiratory diseases in pediatric patients (coefficient: -37.66; 95% confidence interval: -68.17 to -7, 15; $p$ = 0.016).

The distribution behavior of the total number of admissions and hospitalizations diagnosed with pneumonia is quite similar given the proportion of pneumonia cases (over 60%). This predominance of pneumonia cases can also be seen in Fig 2 that shows the distribution of inpatient diagnoses by age group.

Evidence of association of the age group with the diagnosis of hospitalization was found ($p$ <0.001) (Table 3), with highlights for greater proportions of cases of pneumonia in the age groups from three years old than in the age group of zero to two years and all cases bronchiolitis in the range up to two years of age. In the multiple comparisons, no difference was found between the groups of 3 to 5 years old and 11 to 17 years old.

In Fig 3, we see the distribution of the number of hospitalizations per month in the period from 2015 to 2020 for each age group in which the predominance of the age group from a to 2 years is observed very clearly, mainly in the months of seasonal peaks.

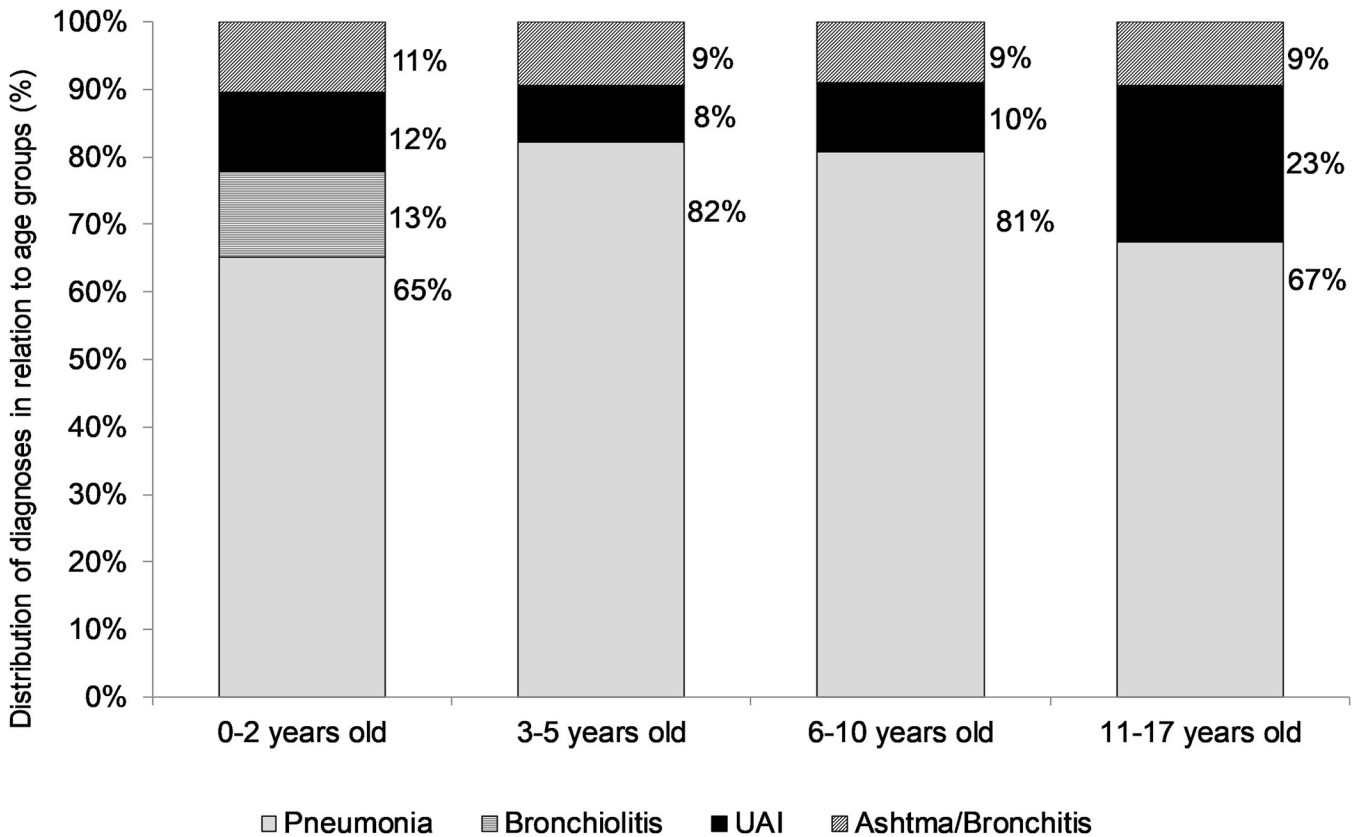

**Fig 2. Inpatient diagnoses by age group of pediatric patients admitted for respiratory diseases in the period from January 2015 to June 2020.**

## Discussion

Our study describes the epidemiological profile of hospitalizations for respiratory diseases in the last 5 years and brings evidence that social isolation has significantly reduced hospitalizations for respiratory diseases in the pediatric population.

**Table 3. Relationship between diagnosis and age groups (n = 2236).**

| Diagnosis | Age groups | | | |
|---|---|---|---|---|
| | **0 to 2 years old** | **3 to 5 years old** | **6 to 10 years old** | **11 to 17 years old** |
| Ashtma/Bronchitis | 133 (10.5%) | 51 (9.4%) | 30 (9.1%) | 9 (9.5%) |
| Bronchiolitis | 163 (12.8%) | 0 (0.0%) | 0 (0.0%) | 0 (0.0%) |
| UAI | 147 (11.6%) | 46 (8.5%) | 34 (10.3%) | 22 (23.2%) |
| Pneumonia | 824 (65.1%) | 446 (82.1%) | 267 (80.7%) | 64 (67.4%) |
| Total | 1267 (100.0%) | 543 (100.0%) | 331 (100.0%) | 95 (100.0%) |
| Global chi-square p-value | <0.001 [1] | | | |
| Multiple comparisons Bonferroni corrected p values | | | | |
| versus 0 to 2 years old | | <0.001 [2] | <0.001 [2] | <0.001 [2] |
| versus 3 to 5 years old | | | >0.99 [2] | 0.002 [2] |
| versus 6 to 10 years old | | | | 0.033 [2] |

p-values for the chi-square test [1] and Fisher's exact test [2]

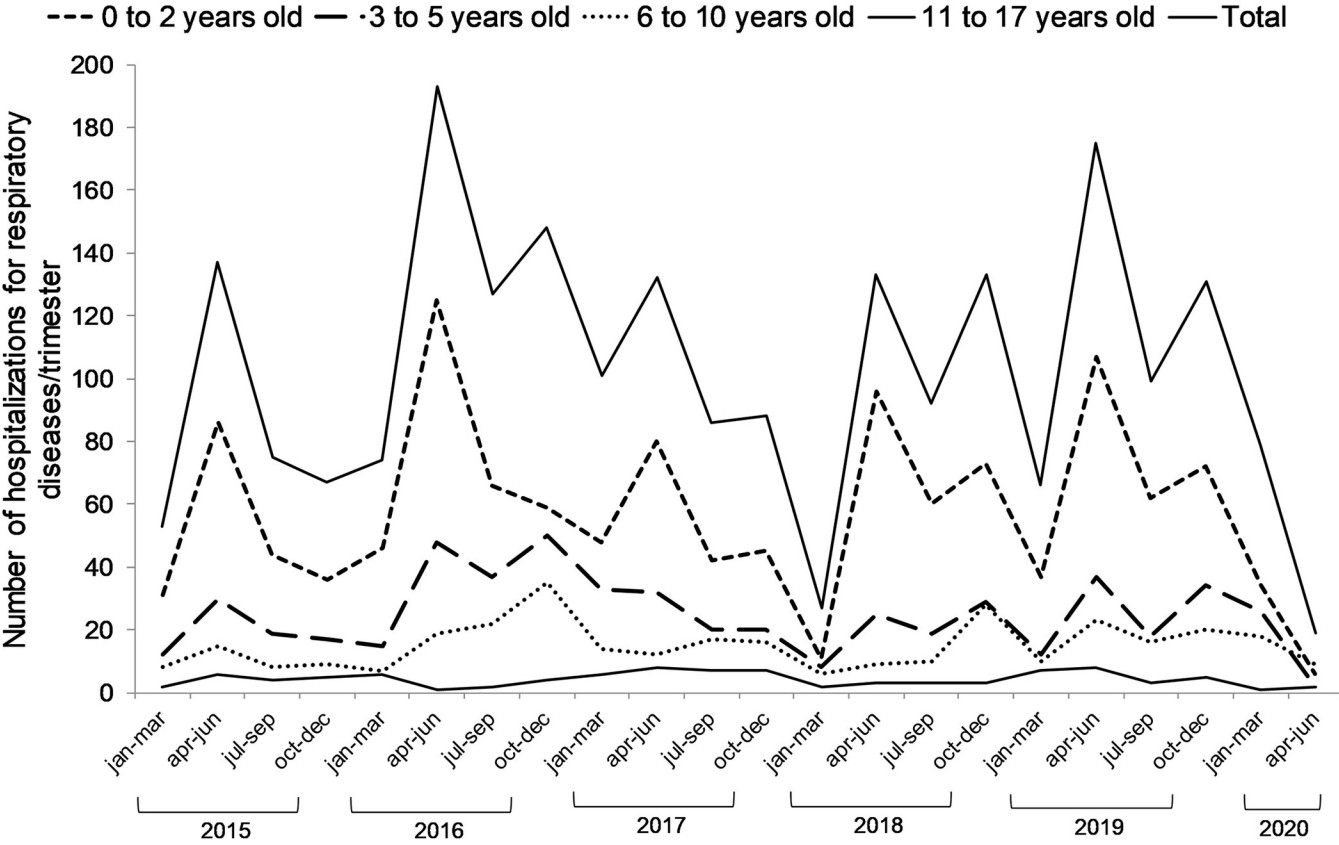

**Fig 3. Distribution of the number of hospitalizations for respiratory diseases in pediatric patients from January 2015 to June 2020, according age group.**

Hospitalizations for respiratory diseases in children and adolescents usually present a distribution pattern dependent on the age group [1, 2] and seasonality [9].

In our study, children under 5 years old represented 81% of hospitalizations and pneumonia were the main diagnoses (over 65%) in all age groups, however, with a higher proportion of pneumonia cases in the age group from three years. Respiratory diseases of viral etiologies (infections of upper airways, bronchiolitis) are cited as being more common in the pediatric population [9, 10], which contradicts our founds. However, other studies corroborate our results and point to pneumonia as a diagnosis in more than 50% of all hospitalizations for respiratory diseases [2, 11, 12].

In our study, when comparing the period with isolation and without isolation, we observed a significant increase in the percentage of hospitalizations in the group of 6 to 10 years. Kuitunen I et al found no difference in the age distribution of patients in periods with and without isolation [13]. We believe that this percentage increase in the number of hospitalizations in the 6 to 10 age group is related to the drastic reduction in the number of hospitalizations during the isolation period. During this period, only 20 patients required hospitalization for respiratory diseases.

Several studies confirm the occurrence of a seasonal peak in respiratory diseases and point to an association with environmental factors such as tobacco exposure, humidity, temperature and an increase in the level of pollutants [14–16]. Sudden changes in temperatures associated with the worsening of the quality of the inspired air are contributing factors for a significant increase in cases of pneumonia, asthma and bronchiolitis [17, 18]. Our results also

demonstrated this seasonality pattern of respiratory diseases in the period from April to June 2015 to 2019, the exception being the year 2020.

Other factors such as the presence of siblings and crowding are also associated with an increased risk of hospitalization for respiratory diseases [12]. Attendance to daycare centers and schools are also identified as contributing factors, not only in the increase in the incidence of respiratory diseases in childhood, but also in the maintenance of the circulation of pathogens [19, 20].

The pandemic caused by COVID-19 in 2020 and the significant reduction in the number of hospitalizations for respiratory diseases in the pediatric population observed by our study demonstrates that, agglomerations and attendance at daycare centers and schools, have a greater contribution share than if could study, as they are difficult to control variables in routine situations.

Another 2 recently published studies confirm our findings and strongly suggest that social distancing and other lockdown strategies are effective to slow down the spreading of respiratory diseases and decreasing the need for hospitalization among children [13, 21].

Since then, the exponential growth in the number of cases, resulting in the collapse of the health system in different countries, has led several governments to adopt control measures to reduce the levels of transmission [22, 23]. These measures included suspending classes at schools and universities, closing companies and commerce and banning events. A study that evaluated the effectiveness of adopting such measures in slowing the growth rates of COVID-19 cases demonstrated that this has a direct association with the time of the epidemic in which they were adopted. In Italy and Spain, control measures were taken at the national level in a final stage of the epidemic, which may have contributed to the high spread of COVID-19 in these countries. In Brazil, the measures adopted prevented the collapse of the health system and appear to have an influence on the growth curve of new infections by COVID-19 [7, 8].

In addition to social isolation measures, the education of the population in relation to the use of masks and hand hygiene were also factors that contributed to curb the circulation of pathogens. Rare were cases of bronchiolitis, highly prevalent diseases at this time of year [24, 25].

Historically, since the Spanish flu, we have not had this spectrum of social isolation including closing schools and daycare centers. This fact transformed the seasonality model of pediatric respiratory diseases. Currently, we only have the question: What will 2021 be like?

## Conclusion

The social isolation measures adopted during the COVID-19 pandemic dramatically interfered with the seasonality of childhood respiratory diseases. This was reflected in the unexpected reduction in the number of hospitalizations in the pediatric population during this period.

## Supporting information

**S1 File.**
(XLSX)

## Author Contributions

**Conceptualization:** Milena Siciliano Nascimento, Cristiane do Prado.

**Formal analysis:** Milena Siciliano Nascimento, Diana Milena Baggio, Linus Pauling Fascina.

**Methodology:** Milena Siciliano Nascimento, Diana Milena Baggio, Cristiane do Prado.

**Writing – original draft:** Milena Siciliano Nascimento, Diana Milena Baggio, Linus Pauling Fascina, Cristiane do Prado.

**Writing – review & editing:** Milena Siciliano Nascimento, Diana Milena Baggio, Linus Pauling Fascina, Cristiane do Prado.

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
