## [Decision Letter · Decision Letter 0]

13 Oct 2020

PONE-D-20-29375

“Impact of social isolation due to COVID-19 on the seasonality of pediatric respiratory diseases”

PLOS ONE

Dear Dr. Nascimento,

Thank you for submitting your manuscript to PLOS ONE. After careful consideration, we feel that it has merit but does not fully meet PLOS ONE’s publication criteria as it currently stands. Therefore, we invite you to submit a revised version of the manuscript that addresses the points raised during the review process.

We look forward to receiving your revised manuscript.

Kind regards,

Brenda M. Morrow, PhD

Academic Editor

PLOS ONE

Journal Requirements:

2. Thank you for including your ethics statement:  "After approval of the project by the Research Ethics Committee, a survey was made of all patients aged 0 to 17 years and 11 months who were hospitalized with primary and secondary diagnosis of respiratory diseases (International Disease Code, 10th Revision: J00 – J99) from January 2015 to July 2020."

Reviewers' comments:

Reviewer's Responses to Questions

**Comments to the Author**

1. Is the manuscript technically sound, and do the data support the conclusions?

Reviewer #1: Partly

Reviewer #2: Yes

2. Has the statistical analysis been performed appropriately and rigorously? 

Reviewer #1: No

Reviewer #2: I Don't Know

3. Have the authors made all data underlying the findings in their manuscript fully available?

Reviewer #1: No

Reviewer #2: No

4. Is the manuscript presented in an intelligible fashion and written in standard English?

Reviewer #1: Yes

Reviewer #2: Yes

5. Review Comments to the Author

Reviewer #1: The current manuscript describes the seasonality of pediatric upper respiratory infections and the impact of the COVID-19 on the incidences. The report is written in understandable English. The topic is current as the pandemic is still going strong globally. The authors however have some difficulties in presenting the results and the discussion in relation to current knowledge remains vague. I have some questions and comments to be addressed by the authors:

1. Abstract. I’d like the authors to comment shortly about the change in the seasonality that was observed. For example: the social isolation guideline decreased majorly pediatric hospitalizations due respiratory infections. As now the conclusion only states that seasonality was interfered.

2. Abstract line 60, correct COVID-10 to COVID-19.

3. Abstract lines 61 and 63. I would suggest the authors to redefine this statement as now it is unclear, and the final message needs to be more precise.

4. Introduction: I would like to hear more about the social restrictions implemented in Brazil to battle covid-19. For example, were day care centers or pre schools closed in the study area?

5. Introduction line 86: the authors aim to report the prevalence of the hospitalizations, however exact numbers are given and no pediatric population size for the area has been given. Incidence would be more suitable in this given context or if population size is unclear then rates should be interpreted.

6. Methods first chapter: The size of the pediatric population in the area could be provided and the profile of the hospital should be mentioned as primary, secondary or tertiary level center.

7. Methods line 103. The exact grouping of the ICD-10 codes into each subcategory should be specified.

8. Methods line 111. The results of these analysis are not shown. Please provide these, as in the results line 146 the authors state evidence of the association was seen… p<0.001, which is not enough.

9. Methods line 120-121. Please provide the version of the R software and the names of the packages that were used.

10. Results table 1, the length of hospital stay is in Portuguese, please translate to English.

11. Results: I’d personally would like the authors to present more results, as now, mostly the results section is only stating what the tables and figures contain, instead of picking results up

12. In order the compare the effect of the lockdown the table 1 could compare the lockdown period to non-lockdown period.

13. The figure 1 is understandable and clear but the resolution of the image is poor and the numbers small

14. The figure 2 is hard to read as the lines are too close to each other. This figure could benefit from. My suggestion is that the authors could consider presenting the figures 1 and 2 together for example as a stacked area chart, where the number and trends of monthly admissions and the reasons for admission could be seen.

15. Results line 137: The adjusting could be described shortly here. Crude report showed xx.x and when adjusted with gender and age the adjusted xxx was xxx. This would benefit the reader majorly. Although I give credit for using confidence intervals here.

16. When adjusting any statistical models the rational behind the selection of the covariates included in the adjusted model should be presented in the methods and explained, why these variates were taken into the model.

17. Discussion: lines 185-187 could be removed.

18. Lines 200 to 203 have no references. I’d suggest the authors to cite for example:

-Seasonal Influenza Activity During the SARS-CoV-2 Outbreak in Japan JAMA 2020 May 19;323(19):1969-1971. doi: 10.1001/jama.2020.6173.

- Does COVID-19 infection impact on the trend of seasonal influenza infection? 11 countries and regions, from 2014 to 2020 Int J Infect Dis 2020 Aug;97:78-80. doi: 10.1016/j.ijid.2020.05.088. Epub 2020 May 31.

- Effect of Social Distancing Due to the COVID-19 Pandemic on the Incidence of Viral Respiratory Tract Infections in Children in Finland During Early 2020 Pediatr Infect Dis J. 2020 Jul 28. doi: 10.1097/INF.0000000000002845. Online ahead of print.

19. The newly published results from other countries on the incidences of respiratory infections during social isolation could be discussed in relation to the results in this study. For example:

- Effect of Social Distancing Due to the COVID-19 Pandemic on the Incidence of Viral Respiratory Tract Infections in Children in Finland During Early 2020 Pediatr Infect Dis J. 2020 Jul 28. doi: 10.1097/INF.0000000000002845. Online ahead of print.

- Social Distancing for COVID-19 and Diagnoses of Other Infectious Diseases in Children Pediatrics. 2020 Sep 2;e2020006460. doi: 10.1542/peds.2020-006460. Online ahead of print.

20. Conclusion is sound and based on the provided results. However the authors could improve the result and the discussion.

In conclusion the authors should address the methodological problems, the issues in presenting the results and how the results are discussed.

Reviewer #2: PLOS ONE manuscript review

Impact of social isolation due to COVID-19 on the seasonality of pediatric respiratory diseases

This is an important paper that highlights the effect of public health strategies instituted to control the spread of COVID-19 globally in 2020. It highlights that these strategies yielded an overall reduction in hospitalisations due to lower respiratory tract infections especially in the under 5 age groups.

My comments below for the authors to consider:-

Abstract:

Rephrase the sentence….. we will only know if there is a change in behaviour observed in 2020, it will also influence the seasonality of 2021 with the continuity of results for……

Methods

Line 94: Patients aged 0 to 17 years and 11 months….. On the results sections, the highest age recruited was 17 years; probably delete the 11 months part on the methodology or just state children under 18 years were recruited.

Line 103 and 10: regarding groupings of the various diagnosis, how was this classification arrived at? Please see my comment below regarding this classification in Table 1.

Results

Table 1:” to” is missing in the rows reading age group

Age group:

0 a 2 years old

3 a 5 years old

6 a 10 years old

11 a 17 years old

Diagnosis

I am not clear why the diagnosis of pneumonia is separate from Bronchopneumonia instead of classifying these two categories under pneumonia.

Also the diagnosis of bronchitis versus bronchiolitis, how were the two diagnoses differentiated especially in the younger children age-group?

It is unclear in the results section the various viral aetiologies depicted under viral pneumonia yet we see this on line 202 in the discussion.

The row that reads …Tempo de internação (dias).. please write this in English

Line 137-that starts…..In the adjusted model,….A table showing the results of this analysis will be useful.

Figure 1: This trend is for all groups. Possible to show the trend for different age groups in Figure1?

Figure 2: Classify pneumonia and Bronchopneumonia together (please see my comments above). Also make the markings on the graph clearer.

Figure 3: see my comments above on the classification of the various diagnosis. Please make the labels of figure 3 clear. Also rephrase line 147 to 149 which is the narrative for figure 3

Discussion

Line 165- again pneumonia and bronchopneumonia terminologies… Please see my comments above regarding these two terminologies

Conclusion: This is well written and summarises the overall take-home message well from this study

6. PLOS authors have the option to publish the peer review history of their article (what does this mean?). If published, this will include your full peer review and any attached files.

Reviewer #1: No

Reviewer #2: **Yes: **Dr Leah N. Githinji

---

## [Author Response · Author response to Decision Letter 0]

28 Oct 2020

São Paulo, Oct 23th 2020

Response to reviewers

Dear Reviewer 1,

Thank you for the revision of our manuscript (PONE-D-20-29375). We appreciated the comments and suggestions, and the manuscript has been discussed or corrected as follows.

Answers are highlighted in red. 

Reviewer #1: The current manuscript describes the seasonality of pediatric upper respiratory infections and the impact of the COVID-19 on the incidences. The report is written in understandable English. The topic is current as the pandemic is still going strong globally. The authors however have some difficulties in presenting the results and the discussion in relation to current knowledge remains vague. I have some questions and comments to be addressed by the authors:

1. Abstract. I’d like the authors to comment shortly about the change in the seasonality that was observed. For example: the social isolation guideline decreased majorly pediatric hospitalizations due respiratory infections. As now the conclusion only states that seasonality was interfered.

We agree with the observation and then we modified the conclusion (LINES 58-61 and 250-253)

2. Abstract line 60, correct COVID-10 to COVID-19.

Correction performed (LINE 59)

3. Abstract lines 61 and 63. I would suggest the authors to redefine this statement as now it is unclear, and the final message needs to be more precise.

We agree with the observation and then we modified the conclusion (LINES 58-61 and 250-253)

4. Introduction: I would like to hear more about the social restrictions implemented in Brazil to battle covid-19. For example, were day care centers or pre schools closed in the study area?

Have been included a paragraph and a reference (8) in the introduction detailing the measures adopted in Brazil (LINES 88-91)

5. Introduction line 86: the authors aim to report the prevalence of the hospitalizations, however exact numbers are given and no pediatric population size for the area has been given. Incidence would be more suitable in this given context or if population size is unclear then rates should be interpreted.

We agree that the term “incidence” is more appropriate within our results. The text has been modified (LINE 92)

6. Methods first chapter: The size of the pediatric population in the area could be provided and the profile of the hospital should be mentioned as primary, secondary or tertiary level center.

Our hospital is a tertiary hospital. The average number of hospitalizations per year in pediatric units is 3000 admissions / year. This information was inserted in the methods section (LINES 100-102)

7. Methods line 103. The exact grouping of the ICD-10 codes into each subcategory should be specified.

We agree with the reviewer's suggestion and we have included the subcategories of the ICD-10 codes in the form of a table to make the grouping that was done clearer (table 1)

8. Methods line 111. The results of these analysis are not shown. Please provide these, as in the results line 146 the authors state evidence of the association was seen… p<0.001, which is not enough.

This question was directed to the statistician who performed the analysis of our study. The text in the methods section was modified with the correction of the analyzes and we added table 3 in the results section (LINES 123-127 and 171-176)

9. Methods line 120-121. Please provide the version of the R software and the names of the packages that were used.

The Version of the R software and the names of the packages that were used were added in the methods section (LINES 135-137).

10. Results table 1, the length of hospital stay is in Portuguese, please translate to English.

The table 1 is now table 2. The term was translated in table 2.

11. Results: I’d personally would like the authors to present more results, as now, mostly the results section is only stating what the tables and figures contain, instead of picking results up

Results have been complemented to make the findings of our study more interesting for the reader. In addition, we changed the figures at the reviewers' suggestion and inserted a new table which also improves our results (LINES 148-157; 165-168; 171-176 and 179-182) 

12. In order the compare the effect of the lockdown the table 1 could compare the lockdown period to non-lockdown period.

Table 1 is now table 2. Comparison analysis was performed between the period with social isolation and without social isolation, as suggested by the reviewer. This new analysis made it possible to find some differences that had not been evidenced. Table 2 was modified and the new results found were described.

13. The figure 1 is understandable and clear but the resolution of the image is poor and the numbers small.

The answer to that question is described below

14. The figure 2 is hard to read as the lines are too close to each other. This figure could benefit from. My suggestion is that the authors could consider presenting the figures 1 and 2 together for example as a stacked area chart, where the number and trends of monthly admissions and the reasons for admission could be seen.

Figures 1 and 2 were unified as suggested by the reviewer. We chose to present the evolution of time in trimesters to make the figure clearer. As in Brazil the months with the highest incidence of respiratory diseases are April, May and June, we believe that this representation was a good alternative.

15. Results line 137: The adjusting could be described shortly here. Crude report showed xx.x and when adjusted with gender and age the adjusted xxx was xxx. This would benefit the reader majorly. Although I give credit for using confidence intervals here.

Answer to questions 15 and 16 are together.

16. When adjusting any statistical models the rational behind the selection of the covariates included in the adjusted model should be presented in the methods and explained, why these variates were taken into the model.

This question was directed to the statistical professional who performed the analysis of our study and the following answer was sent.

The only variables in this model were: the number of hospitalizations, the referring month and whether or not it was a period of isolation. There was no adjustment based on any other variable. In this model, we are working with grouped data, so gender and age of individuals were not considered.

To make our analysis clearer the word “adjusting” was removed and we emphasized the name of the test that was used in the “ARIMA” model both in the methods section and in the description of the results (LINES 129-130 and 161)

17. Discussion: lines 185-187 could be removed.

We agreed with the reviewer that this information was redundant in the discussion. Lines 185-187 have been removed.

18. Lines 200 to 203 have no references. I’d suggest the authors to cite for example:

-Seasonal Influenza Activity During the SARS-CoV-2 Outbreak in Japan JAMA 2020 May 19;323(19):1969-1971. doi: 10.1001/jama.2020.6173.

- Does COVID-19 infection impact on the trend of seasonal influenza infection? 11 countries and regions, from 2014 to 2020 Int J Infect Dis 2020 Aug;97:78-80. doi: 10.1016/j.ijid.2020.05.088. Epub 2020 May 31.

- Effect of Social Distancing Due to the COVID-19 Pandemic on the Incidence of Viral Respiratory Tract Infections in Children in Finland During Early 2020 Pediatr Infect Dis J. 2020 Jul 28. doi: 10.1097/INF.0000000000002845. Online ahead of print.

We appreciate the suggestions. References 24 and 25 have been added (LINES 241-244)

19. The newly published results from other countries on the incidences of respiratory infections during social isolation could be discussed in relation to the results in this study. For example:

- Effect of Social Distancing Due to the COVID-19 Pandemic on the Incidence of Viral Respiratory Tract Infections in Children in Finland During Early 2020 Pediatr Infect Dis J. 2020 Jul 28. doi: 10.1097/INF.0000000000002845. Online ahead of print.

- Social Distancing for COVID-19 and Diagnoses of Other Infectious Diseases in Children Pediatrics. 2020 Sep 2;e2020006460. doi: 10.1542/peds.2020-006460. Online ahead of print.

We added 2 new paragraphs to the discussion with the references suggested by the reviewer (LINES 199-206 and 225-228) (References 13 and 21)

20. Conclusion is sound and based on the provided results. However the authors could improve the result and the discussion.

The “result” and “discussion” sessions had their texts supplemented and / or modified 

In conclusion the authors should address the methodological problems, the issues in presenting the results and how the results are discussed.

Dear Reviewer 2,

Thank you for the revision of our manuscript (PONE-D-20-29375). We appreciated the comments and suggestions, and the manuscript has been discussed or corrected as follows.

Answers are highlighted in bold. Parts of the text that have been modified are in Italic.

Reviewer #2: PLOS ONE manuscript review

 Impact of social isolation due to COVID-19 on the seasonality of pediatric respiratory diseases

 This is an important paper that highlights the effect of public health strategies instituted to control the spread of COVID-19 globally in 2020. It highlights that these strategies yielded an overall reduction in hospitalisations due to lower respiratory tract infections especially in the under 5 age groups.

 My comments below for the authors to consider:-

 Abstract:

 Rephrase the sentence….. we will only know if there is a change in behaviour observed in 2020, it will also influence the seasonality of 2021 with the continuity of results for……

We agree with the observation and then we modified the conclusion (LINES 58-61 and 250-253)

Methods

 Line 94: Patients aged 0 to 17 years and 11 months….. On the results sections, the highest age recruited was 17 years; probably delete the 11 months part on the methodology or just state children under 18 years were recruited.

“And 11 months” removed from the text both in the abstract and in the methods

 Line 103 and 10: regarding groupings of the various diagnosis, how was this classification arrived at? Please see my comment below regarding this classification in Table 1.

The grouping was performed through the subcategories of the ICD-10 codes. To make it clearer, we include the subcategories of the ICD-10 codes in the form of a table (table 1) to clarify the grouping that was carried out and we grouped bronchopneumonia and pneumonia into a single category, as suggested by the reviewer. After this new categorization, a new statistical analysis was performed,

Results

 Table 1:” to” is missing in the rows reading age group

 Age group:

 0 a 2 years old

 3 a 5 years old

 6 a 10 years old

 11 a 17 years old

The table 1 is now table 2. The terms have been corrected in the table 2 

Diagnosis

 I am not clear why the diagnosis of pneumonia is separate from Bronchopneumonia instead of classifying these two categories under pneumonia.

We grouped bronchopneumonia and pneumonia in one category, as suggested by the reviewer. After this new categorization, a new statistical analysis was performed.

Also the diagnosis of bronchitis versus bronchiolitis, how were the two diagnoses differentiated especially in the younger children age-group?

We agree with the reviewer that there are no marked clinical differences between the two diagnoses (bronchitis and brochiolitis) mainly in younger children. However, the diagnosis of bronchitis was closely associated with the secondary diagnosis of asthma. For this reason, there was a grouping of these 2 diagnoses. This explanation was inserted in the methods section. (LINES 115-117)

It is unclear in the results section the various viral a etiologies depicted under viral pneumonia yet we see this on line 202 in the discussion.

We agree with the reviewer that there is no description of viral pathogens in the results. These data have not really been collected. However, the authors' idea was to show the cases of bronchiolitis, a disease caused mainly by the respiratory syncytial virus. The terms have been replaced in this paragraph (LINES 241-244).

The row that reads …Tempo de internação (dias).. please write this in English

The table 1 is now table 2. The term was translated in table 2.

Line 137-that starts…..In the adjusted model,….A table showing the results of this analysis will be useful.

This question was directed to the statistical professional who performed the analysis of our study and the following answer was sent.

 The only variables in this model were: the number of hospitalizations, the referring month and whether or not it was a period of isolation. There was no adjustment based on any other variable. In this model, we are working with grouped data, so gender and age of individuals were not considered. 

To make our analysis clearer the word “adjusting” was removed and we emphasized the name of the test that was used in the “ARIMA” model both in the methods section and in the description of the results (LINES 129-130 and 161)

Figure 1: This trend is for all groups. Possible to show the trend for different age groups in Figure1?

Figure 1 and figure 2 were unified at the request of reviewer 1. To make the visualization of the figures clearer for the reader, we chose to bring a trend for different age groups in a new figure (Figure 3)

Figure 2: Classify pneumonia and Bronchopneumonia together (please see my comments above). Also make the markings on the graph clearer.

We grouped bronchopneumonia and pneumonia in a single category as suggested by the reviewer and the figure was modified.

Figure 3: see my comments above on the classification of the various diagnosis. Please make the labels of figure 3 clear. Also rephrase line 147 to 149 which is the narrative for figure 3

After classifying Bronchopneumonia and pneumonia together, the statistical analysis was redone. We included a new table (table 3) for presenting the results and the text was supplemented (LINES 171-176)

Discussion

Line 165- again pneumonia and bronchopneumonia terminologies… Please see my comments above regarding these two terminologies

We grouped bronchopneumonia and pneumonia in a single category The terms have been corrected.

 Conclusion: This is well written and summarises the overall take-home message well from this study

---

## [Decision Letter · Decision Letter 1]

26 Nov 2020

“Impact of social isolation due to COVID-19 on the seasonality of pediatric respiratory diseases”

PONE-D-20-29375R1

Dear Dr. Nascimento,

We’re pleased to inform you that your manuscript has been judged scientifically suitable for publication and will be formally accepted for publication once it meets all outstanding technical requirements.

Kind regards,

Brenda M. Morrow, PhD

Academic Editor

PLOS ONE

Additional Editor Comments (optional):

Reviewers' comments:

Reviewer's Responses to Questions

**Comments to the Author**

1. If the authors have adequately addressed your comments raised in a previous round of review and you feel that this manuscript is now acceptable for publication, you may indicate that here to bypass the “Comments to the Author” section, enter your conflict of interest statement in the “Confidential to Editor” section, and submit your "Accept" recommendation.

Reviewer #1: All comments have been addressed

Reviewer #2: All comments have been addressed

2. Is the manuscript technically sound, and do the data support the conclusions?

Reviewer #1: Yes

Reviewer #2: (No Response)

3. Has the statistical analysis been performed appropriately and rigorously? 

Reviewer #1: Yes

Reviewer #2: (No Response)

4. Have the authors made all data underlying the findings in their manuscript fully available?

Reviewer #1: Yes

Reviewer #2: (No Response)

5. Is the manuscript presented in an intelligible fashion and written in standard English?

Reviewer #1: Yes

Reviewer #2: (No Response)

6. Review Comments to the Author

Reviewer #1: I would like to thank the authors for answering all of my concerns. This paper has had major improvement. I have no further questions or comments regarding this manuscript.

Reviewer #2: All comments have been addressed.

You may correct the small typo on line 99 (provide) and missing comma after URI on line 112

7. PLOS authors have the option to publish the peer review history of their article (what does this mean?). If published, this will include your full peer review and any attached files.

Reviewer #1: No

Reviewer #2: No

---

## [Editor Report · Acceptance letter]

4 Dec 2020

PONE-D-20-29375R1 

Impact of social isolation due to COVID-19 on the seasonality of pediatric respiratory diseases. 

Dear Dr. Nascimento:

I'm pleased to inform you that your manuscript has been deemed suitable for publication in PLOS ONE. Congratulations! Your manuscript is now with our production department. 

Kind regards, 

on behalf of

Professor Brenda M. Morrow 

Academic Editor

PLOS ONE